# Young Adult Cancer Care Partners: A Theoretical Description of an Emerging Population with Unique Needs

**DOI:** 10.3390/ijerph20176646

**Published:** 2023-08-25

**Authors:** Echo L. Warner, Megan Hebdon, Djin L. Tay, Keely Smith, Anna Welling, Jiayun Xu

**Affiliations:** 1Huntsman Cancer Institute, University of Utah, Salt Lake City, UT 84112, USA; 2College of Nursing, University of Utah, Salt Lake City, UT 84112, USA; 3School of Nursing, University of Texas at Austin, Austin, TX 78712, USA; 4College of Nursing, University of Arizona, Tucson AZ 85721, USA; 5Purdue Center for Families, West Lafayette, IN 47907, USA; 6College of Nursing, Purdue University, West Lafayette, IN 47907, USA

**Keywords:** cancer, caregiver, young adult, social network, social support, life-course, developmental stage, narrative review

## Abstract

As the U.S. population’s demographics shift, young U.S. adults are increasingly engaged in informal caregiving for aging generations. Yet, there is little research on the unique experiences and needs of young adults who take on caregiving roles for adult cancer patients. Herein we demonstrate through a theoretical description that young adult cancer care partners deserve distinct recognition in the cancer control continuum given the psychological, physical, financial, and social features unique to their cancer experience.

## 1. Introduction

Young adult cancer caregivers (YACCs), also known as young cancer caregivers, millennial caregivers, or young caregivers, have been defined in various ways. If we consider the definitions in our own research on this population, along with definitions from national organizations such as the National Cancer Institute, American Association for Retired People (AARP), and National Caregiving Alliance, we can define YACCs as individuals aged 18–39 years who are engaged in informal care for a cancer patient (see Table 1). YACCs typically care for a parent or grandparent, balance multiple caregiving responsibilities, work in an hourly job for at least 30 h a week, and spend at least 20 h a week in providing care [1,2]. Here, we will consider them as caregivers or care partners.

## 2. Background

Young adults, those aged 18–39 years, are increasingly taking on caregiving roles for similarly aged and older adults. In 1997 in the U.S., 22.3% of care partners were under the age of 35; in 2020, this number was 24% [4,5]. An estimated 20.9 million young adults serve as care partners in the U.S., with 1.46 million providing care to a cancer patient [6].

Caring for a cancer patient is not a normative task for YACCs; in comparison with older care partners, it places them at greater risk for caregiver stress. Unforeseen caregiving can disrupt YACCs’ well-being and prompt fear of abandonment and loss of the patient, as it may for older care partners [7,8,9]. However, YACCs experience such challenges at a time of life when they may be less established in their careers, with fewer resources and less stable social support [10,11]. Compared with older cancer partners, YACCs report greater stress and depression during the first 6 months of caregiving, although those who have better relationships with their cancer patients do have less stress and depression during this initial transition period [9]. However, YACCs are more likely than older care partners to express high levels of unmet needs within the first 6 months of diagnosis, and they commonly report unmet information needs related to their caregiving role [12].

Key developmental milestones (e.g., emotional independence and financial stability) may be delayed when YACCs set their personal aspirations aside to care for a family member or friend with cancer [13,14]. Although caregiving can elicit feelings of closeness and a sense of fulfillment, it can limit YACCs’ social engagement and lead to isolation and depression [13,15]. For YACCs, becoming a care partner requires integrating new duties with existing responsibilities (e.g., caring for young children, completing education), and this can have life-long impacts on relationships, financial attainment, and career development. Social support during the first 6 months of caregiving can help to ease the burden of an individual’s new caregiving role [16], but it is not always available or sustained during this time. Social media networks offer a potential source of social support to shield YACCs from becoming overwhelmed in their new caregiving roles [4].

In this study, we consider YACCs in the context of the cancer family caregiving experience, a theoretical model that has driven caregiving research over the past decade [6]. At the same time, we use a life course perspective as a developmental lens to show how the young adult cancer caregiving experience influences life events. Young adult cancer caregiving affects three critical areas of young adult development along the life course: family and relationships, financial well-being, and health outcomes. There is a need to improve support for YACCs across the cancer control continuum and enhance equitable cancer support services for cancer care partners of all ages. Increasing evidence shows that YACCs are a unique caregiving population with unique needs and experiences pertaining to family and relationships, financial hardship, health outcomes, and social support.

## 3. The Cancer Family Caregiving Experience

The cancer family caregiving experience model posits three main components of informal caregiving: contextual factors, the stress process, and the cancer trajectory [6]. In this model, primary stressors initiate the stress process and include decreased social support, poor coping skills, and patient-related factors including disease site, stage, prognosis, and duration; symptoms; and functional dependency [6]. As they address these complex primary stressors, YACCs must meet the demands of caregiving, which include managing complex medications, navigating the healthcare system, and reducing symptom burden [14,17]. Secondary stressors arise as these demands begin to affect other aspects of life. For YACCs, secondary stressors can arise as caregiving demands begin to affect their social interactions, romantic relationships, parenting responsibilities, employment, and education. However, the majority of research on secondary stressors has focused on older adult care partners, not on how caregiving demands may be different for YACCs.

As primary and secondary stressors begin to weigh on YACCs, they negotiate how to cope with each stressor [18]. This negotiation involves the YACC’s cognitive appraisal of the stressor, and cognitive appraisals can be positive (e.g., with confidence or preparedness) or negative (e.g., with uncertainty or hopelessness) [19]. When the YACC appraises stressors related to caregiving, this process is not just individual; it may be heavily influenced by social networks. For example, using social media, YACCs may try to compare their experience with their peers’ experiences. As YACCs seek support through social media, their comparisons with others’ situations can offer opportunities to appraise their own caregiving roles, their responsibilities to their families and work, their social lives, and their goals. Such appraisals may benefit individuals by connecting them with other young adult care partners who can validate their experiences, or they may have negative consequences if YACCs appraise their situations as less optimal in comparison with the situations of others in their social network. This influence of social networks has not yet been explored in the literature.

Finally, stress can influence the YACCs’ health and well-being. Positive outcomes of caregiving include, for example, increased life satisfaction and sense of meaning; but among cancer care partners, negative outcomes such as depression and emotional disturbance are common [18]. The literature suggests that YACCs experience more negative mental health outcomes (e.g., depression, anxiety) than do older care partners [9,12]. The cancer family caregiving experience model provides a framework for comprehending the supportive care needs of YACCs at different stages of the stress process and cancer trajectory within a broader sociocultural context. Cancer caregiving experiences during young adulthood have been under-evaluated, and much of what is known about this age group has been extrapolated from studies of older care partners without considering developmental age. Furthermore, the normative social development and emergent social issues that arise during young adulthood likely differ greatly from those arising among older adults. Nevertheless, although the cancer family caregiving experience model is foundational for the study of cancer caregiving, it does not fully encapsulate YACCs’ supportive care needs during their initial period of transition to the caregiving role. The heterogeneity of social development during young adulthood is an important but previously undescribed feature of the model [20]. This necessitates the use of a life course perspective.

## 4. Life Course Perspective

YACCs enter their caregiving role at a unique stage of psychological development. Generally, at least in economically developed countries, young people between the ages of 15 to 39 years finish school, transition from their childhood homes, enter and establish themselves in the workforce, form intimate and long-term emotional and sexual relationships, and start families. Key developmental events experienced by these young adults do vary (see Table 2) [4,7,10,11,21,22,23]. For example, emerging adults (aged 18–25 years) are often in the process of leaving their childhood home, obtaining higher education, career training, establishing their own world views, and creating social connections independent from those of childhood [10]. The transition from emerging adulthood to young adulthood (approximately ages 26–39 years) is not as definitively marked as the transition from adolescence to emerging adulthood, given the increasing heterogeneity of individuals’ vocations and relationships. By their late 20s or early 30s, most young adults have become independent from their childhood caregivers by establishing financial self-sufficiency, accepting responsibility for their actions, developing their own personal beliefs, and establishing adult relationships with their parents [22].

However, for YACCs, taking on a cancer caregiving role to support a cancer patient through diagnosis, treatment, and subsequent late effects can disrupt social maturation—the process with which young people develop their views of self, as well as the social cognition, awareness, and emotional regulation that guide them through life [7]. Furthermore, young adults who engage in caregiving for an individual who was once their parent or caretaker may find themselves experiencing role reversal during their young and middle adulthood. Role reversal can be rewarding, but it can also present difficulties and conflict. Thus, cancer-related disruptions can have deleterious implications for the life-long health and well-being of YACCs.

## 5. Family and Relationships

Social connections are critically important in young adulthood, because they form the social support on which many care partners rely along the cancer trajectory and over their life course [10,11,21]. Social connections include friendships and online social networks: close friendships, familial relations, and intimate partnerships. The establishment and maintenance of these relationships is complicated when a young adult becomes a cancer caregiver. On the one hand, YACCs may leverage their social connections to form a team or network of individuals who contribute to their caregiving responsibilities overall [24]. On the other hand, YACCs may become isolated and feel alone if they do not have robust social networks to help them fulfill their caregiving responsibilities [25]. For isolated YACCs, seeking connections with other young adults may help to normalize the caregiving experience and provide emotional and informational support. YACCs often use social media networks to find peers to form connections and find support [21]. However, many YACCs may not yet have developed strong relationships with other adults during their life course by the time they become engaged in cancer caregiving. They may not have had the time to develop and nurture the social relationships that can provide a foundation for strong social support during cancer care. Cancer caregiving’s disruptions to social connectedness likely have differential, longer lasting negative impacts among YACCs than among older adult cancer care partners, who have often developed a lifetime of social relationships that provide support during cancer experiences.

The increased flexibility in gender norms within younger generations may suggest increasing gender equality in caregiving burden. For example, more young men are taking on caregiving roles than ever before [3], while simultaneously more young women work outside the home, meaning that, the caregiving role historically taken on by women may no longer be as gendered as in the past [4,22]. In addition to less gendered norms of caregiving roles, YACCs’ gender and sexuality may be affected by caregiving during young adulthood. YACC have described their caregiving interrupting their ability to date and explore intimate relationships. Further, young adults who have survived cancer have higher rates of divorce than individuals without a cancer history [23]. While we are aware of no prior studies on the area, cancer caregiving at a young age, particularly for a spouse or intimate partner, may negatively impact these relationships.

## 6. Financial Well-Being, Hardship, Toxicity

In the 2020 AARP national survey of family care partners, an estimated 61% of family care partners were employed outside of the home [4]. To manage their caregiving and work responsibilities, employed care partners must often make work accommodations such as going in late, decreasing work hours, turning down promotions, and accepting the risk of performance warnings [4]. Although some work accommodations are benign, others can be detrimental to the employed care partner’s career progression and long-term career trajectory. Employed YACCs who are starting their careers may be disproportionally affected by the challenge of caregiving while working. For instance, changes to one’s work schedule and turning down job opportunities because of caregiving conflicts can decrease or eliminate opportunities for professional growth and development. In fact, half of YACCs have reported that their career or job prospects have been negatively affected by their caregiving responsibilities [22].

Because employed YACCs are only beginning their careers and have not had time to establish themselves professionally, they may be more hesitant to disclose caregiving responsibilities to supervisors. Although a supervisor’s support is important, employed YACCs may not feel comfortable about disclosing their caregiving situations because they may not yet have a strong work history with their employers [3,26]. Without supervisors’ awareness and support, employed YACCs place themselves at a disadvantage by not having access to resources available for caregiving. Additionally, YACCs may find it difficult to hold their jobs if employers are unaware of the reasons for their work disruptions. Severe warnings, salary cuts, or demotions can be consequences of work accommodations for caregiving.

Like all care partners, YACCs spend an average of USD 6800 per year of their own incomes on caregiving [22]. Other sources involved in paying for caregiving-related expenses include withdrawals from savings accounts and reductions in retirement savings. Compared with middle- and older-aged adults, YACCs typically have fewer savings and less money in retirement funds to use toward caregiving, and the use of their current income limits the money they can save or contribute for retirement. The inability to accrue wealth, or a reduction in savings overall, can have long-lasting impacts on YACCs’ financial security and may increase the likelihood that they may have to rely on public assistance in the future.

## 7. Care Partner Health Outcomes

The health outcomes for young adult care partners are not fully understood (see Table 3). During 2014–2017, for example, young adults experienced increasing rates of hypertension, type II DM, and high cholesterol [27]. Young adults are currently less healthy overall than previous generations were at the same age [27], which is concerning given the strain that caregiving can place on the physical health and well-being of care partners. Greater physical strain has been reported in other young adult caregiving groups including caregivers who provide care for young children and aging parents or grandparents, known as sandwich caregivers (18%), and mental health care partners (22%) [17]. Additionally, 20–26% of caregivers from these two groups have reported that caregiving has worsened their health [22,28]. Compared with older generations, younger generations of adult caregivers may enter older adulthood with worse overall health due to the strain of caregiving and changing demographics [22,27].

Today, rates of mental and emotional health conditions such as depression and substance abuse are higher for young adults than they were for previous generations [29]. Six of the top 10 health conditions faced by young adults include behavioral health [27]. Emerging adult care partners (ages 18–25) have greater mental distress, depression, and anxiety than do non-care partners [8,9]. Additionally, from 2016 to 2020, the number of young adult caregivers (YACs) who reported that caregiving was moderately to highly stressful increased [4,28]. The type of caregiving may impact the experience of mental and emotional distress, with young adult mental health care partners reporting greater levels of emotional stress [22]. Additional considerations for mental and emotional health in YACs include intersectional issues with employment and access to health care; greater sexual, gender, and racial diversity, with associated bias and structural racism; and overall caregiving burden related to time and resources that is not well-defined in this group [22].

Three of the top 10 health issues in young adults include substance use (alcohol, tobacco, and other substances) [27]. In a study of Behavioral Risk Factor Surveillance System data from YACs (aged 18–25 years), rates of tobacco use were higher in care partners than in non-care partners, but there were no differences in alcohol or e-cigarette use [9]. Overall, alcohol and marijuana are the most frequently used substances in young adults aged 18–25 [15]. Given the risk-taking behaviors and social patterns among alcohol and other substance users, recreational substance use or use for coping in YACs should be at the forefront of research to support their health and well-being [27].

In YACs, there are also reported benefits to caregiving: greater self-esteem, coping skills, and positive feelings about caregiving [8,12]. The health impact of these benefits has not been fully explored in YACCs. Emphasizing the perceived benefits of caregiving may be one way to help support YACCs in their caregiving roles.

## 8. Supportive Care and Palliative Care Needs along the Disease Trajectory

Supportive care is targeted at improving quality of life, and it can include symptom management, psychosocial support of patients and care partners, and decision-making support along the cancer trajectory. Palliative care, which is not intended to cure disease, supports the needs of patients and care partners during serious illness until the end of life. There is increasing recognition of the developmental needs of young adults with cancer. However, despite greater unmet psychosocial, informational, and financial needs among YACCs than among older care partners, few studies have examined how supportive and palliative care needs of younger care partners may differ [14,17].

Cancer care partners often assist with hands-on tasks such as activities of daily living, medication administration, and coordination of care [14,17]. Younger age has consistently been associated with cognitive appraisals of greater caregiving burden during active cancer treatment, as well as with poorer coping in caregiving of those with serious illnesses [16,30]. Although cancer caregiving is shorter and more episodic than other forms of caregiving, cancer care partners provide more hours of care, greater assistance with activities of daily living, and greater care coordination than do care partners of those with other diseases and disorders, and they experience greater caregiving burden [17]. YACCs may be at risk for greater psychological distress and death anxiety, which may be related to relational contexts such as attachment styles and histories of family conflict [30,31,32,33,34]. Along with the financial, economic, and developmental needs of YACCs, their additional caregiving responsibilities make access to supportive and palliative care even more critical for those who care for advanced cancer patients.

### 8.1. Geographic Considerations

Current changes in U.S. demographics include lower marriage rates, lower birth rates, and regional and international migration, which affect the availability of care partners for those with a serious illness [35,36]. Distance caregiving is relatively understudied, but approximately 15% to 20% of care is provided by distance care partners who are often patients’ children [37]. Cancer care partners who live far away from the patient may nevertheless participate in periodic direct care, care coordination, and/or the financial burdens of care [18,38]. Care partners geographically distanced from advanced cancer patients may be more vulnerable to anxiety and stress, and they are also less likely to receive formal supportive care services [20,38,39].

### 8.2. Decision Making

Because they often lack experience with chronic and severe diseases and have limited prior exposure to the health care system, YACCs may require greater support for decisions about treatment [40,41]. This can be further complicated by birth order, particularly in cultures where older children are expected to bear the burden of decision making [42]. End-of-life care often involves decisions, with one in eight care partners reporting additional support and resources for making medical decisions regarding serious illness [4]. Given their lack of experience with severe illness and mortality [7], younger care partners likely have greater needs for navigating decision making and gathering information and support for the paperwork involved in documentation of end-of-life wishes or other medical orders [20,43]. When young adults have yet to establish their own identities and beliefs [10,11], this may also negatively affect their ability as surrogate decision makers to predict patients’ end-of-life wishes, another core aspect of supportive care.

## 9. Research and Clinical Priorities

There is a clear need to prioritize clinical and research directives to address YACCs’ unique needs along the cancer control continuum. The cancer family caregiving experience model can enable researchers and clinicians to better understand and support this demographic. YACCs face challenges along their developmental trajectories that differ from those for other caregiving groups. YACCS do not just navigate their roles as care partners; they are maneuvering their ways through major life events such as family building and establishing their careers [7], both of which have significant impacts on their long-term financial well-being [4].

YACCs are a diverse group with intersectional backgrounds based on culture, race, socioeconomic status, education, and sexual and gender identity. Further research must be conducted to build adaptable support systems that meet these caregivers’ needs. YACCs may benefit, for example, from online resources. Although cancer caregiving tends to be shorter than other forms of caregiving [17], it has negative impacts on care partners’ physical and mental health [22,28]. Individualized supportive care systems for YACCs should recognize the benefits of caregiving such as coping skills and self-esteem [8,12], while simultaneously addressing caregiving’s adverse impacts on health. Future priorities for research to characterize the impact of young adult cancer caregiving include, but are not limited to, the impact of caregiving on the developmental life course trajectory, family and relationships (e.g., marriage/divorce, gender norms and equity in caregiving, childbearing), financial outcomes related to caregiving as a young adult and its impact on employment, health insurance, and lifetime earning/debt, health outcomes related to caregiving that may be amplified among young adults such as use of alcohol and other substances, negative (depression, anxiety, stress) and positive (resilience, independence) psychological outcomes of caregiving, geographic considerations for long-distance caregiving, and supportive and palliative care needs among YACCs who may be less familiar with severe illness. Defining the unique clinical needs and research priorities for YACCs will require iterative discussion to ensure that this growing population thrives along their caregiving journey.

## 10. Summary

YACCs are a growing population of care partners with unique social, developmental, and financial needs. Formal recognition of this demographic of care partners in the health care system will help to prioritize their needs, promote an understanding of how caregiving affects them over the short and long term, and inform the development of resources that better address their needs. Advancing policy for paid family leave, paid caregiver leave, employment and health insurance protections, and childcare support for young caregivers may support YACCs in their pursuit of personal, professional, financial, and care partner goals is critical to their overall well-being.

## Figures and Tables

**Table 1 ijerph-20-06646-t001:** Definitions.

Terms	Description
YACC, millennial cancer caregiver, young cancer caregiver	Young adults engaged in providing care to an adult cancer patient who is over age 18. Care partners are typically aged 18–39 years, reflecting the National Cancer Institute’s definition of young adult cancer patients as aged 15–39 years [3].
YAC, millennial caregiver	Young adult care partner; generally, young adults who are care partners for any disease (loosely defined as ages 18–39 years)

**Table 2 ijerph-20-06646-t002:** Social issues and milestones for YACCs by age [7].

Age	Significant Social Relationships	Specific Social Issues/Milestones Related to Cancer Caregiving
Emerging adulthood (18–25 years)	Friends, school and work peers, parents, family	Interrupted social development and peer identification Educational achievement interrupted Delayed transition to independent living Delays or gaps in higher education Interruption and gaps in employment Disruptions in adequate health insurance coverage Hindered achievement of financial independence Difficulty developing or maintaining intimate relationships Delayed or interrupted establishment of financial independence Geographic distance from patient and/or home
Young adulthood (26–39 years)	Partners, family, friends, school and work peers	
Middle adulthood (40–64 years)	Partners, children, family, friends, work peers	
Older adulthood (65+)	Partners, children, grandchildren, family, friends	

Note: YACC, young adult cancer care partner.

**Table 3 ijerph-20-06646-t003:** Health outcomes of young adult cancer caregiving.

Domains	Outcomes
Morbidity/mortality	One fifth of sandwich care partners reported high physical strain, and 20% reported that caregiving made health worse [28]. Top 10 physical health conditions in millennials included hypertension, type II DM, high cholesterol, and inflammatory bowel disease. Millennials were less healthy than GenXers at the same age [27]. 22% of millennial mental health care partners reported physical strain, and 26% reported poorer health due to caregiving [22].
Alcohol/substance use	BRFSS data for adults aged 18–25; care partners had higher rates of cigarette smoking than did non-care partners; no difference in e-cig or alcohol use [9]. 13.5% of young adults aged 18–25 drank alcohol an average of 4.1 drinks per day on 7 days per month, and 9.0% used marijuana [15].
Mental health/psych	Emerging adults aged 18–25; care partners had higher frequent mental distress than did non-care partners [9]. Emerging adult care partners aged 18–25 had higher rates of depression and anxiety versus non-care partners [8]. One third of sandwich care partners reported high emotional stress [28]. Caregiving was moderately to highly emotionally stressful for millennial care partners [4]. Six of the top 10 conditions affecting millennial health were behavioral, including depression and alcohol and substance use disorders [27]. 30% of millennial care partners experienced emotional stress related to caregiving of non-mental health related issues, 45% of those caring for those with mental health related issues experienced distress [22].
Positives	Past and current emerging adult care partners aged 18–25 reported greater self-esteem and coping than did non-care partners, although results were nonsignificant [8]. Young adult care partners (aged 21–40) reported greater positive feelings than did middle-aged or older adult care partners [12].

## Data Availability

No new data were created or analyzed in this study. Data sharing is not applicable to this article.

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
