# Peer review of "Young Adult Cancer Care Partners: A Theoretical Description of an Emerging Population with Unique Needs"

_ijerph, 2023, doi:10.3390/ijerph20176646_

Round 1

Reviewer 1 Report

This paper provides a very thoughtful review of the literature related to the challenges and needs of young adult cancer care partners.  As the authors point out, upwards of one-fourth of care partners in the US are young adults. While the needs of care partners more generally have received increasing attention in recent years, the unique needs of young adults who take on this role merit further consideration. The authors provide a thorough overview of the relevant literature and, based on this, argue that YACCs receive formal recognition as a distinct care provider group.  This paper represents a valuable contribution to the literature that can be leveraged by those seeking to conduct research or policy related work in this area.  

Overall, the paper is very well organized and clearly written.  Areas where clarity could be improved are detailed below:

Table 2:  Only parents are listed as a significant social relationship for the emerging adulthood group, vs. family for the other age groups.  Is there a reason for excluding the broader family category here?

Para spanning lines 165-172 re gender roles and sexuality: The links between gender, sexuality, caregiving for intimate partners and divorce among cancer survivors are muddy.  Suggest reworking this para to make more cohesive and the potential links between findings from prior literature more clear.

Line 200:  More appropriate to indicate that these impacts MAY increase the likelihood that YACCs will have to rely on public assistance in the future

Line 208:  Consider adding the word other -- Greater physical strain has been reported in OTHER young adult caregiving groups including...

Line 209: The term sandwich care partner is introduced here and occurs elsewhere in the manuscript but it is not defined.  This will be an unfamiliar term for some and should be clearly defined.

Table 3:
Outcomes described for morbidity/mortality are depicted as percentages. Outcomes described for alcohol/substance use are numbers. It is hard to have a sense of the magnitude of issues when only numbers are presented. Is 4.8 million adults a large proportion of young adults? What about 3.2 million? 

Line 216 vs Table 3:
Line 216 notes that 6 of the 10 top health conditions faced by young adults include behavioral health. 
Table 3 indicates that the top 10 conditions affecting millennial health were behavioral.
Same reference provided for both statements.  Please revise for accuracy and consistency.

Sections 9 and 10 (Research & Clinical Priorities and Summary)
Specific suggestions regarding the types of studies that may provide the most value (section 9) and the types of policy activities that could be undertaken to support YACC would strengthen the paper.

Author Response

This paper provides a very thoughtful review of the literature related to the challenges and needs of young adult cancer care partners.  As the authors point out, upwards of one-fourth of care partners in the US are young adults. While the needs of care partners more generally have received increasing attention in recent years, the unique needs of young adults who take on this role merit further consideration. The authors provide a thorough overview of the relevant literature and, based on this, argue that YACCs receive formal recognition as a distinct care provider group.  This paper represents a valuable contribution to the literature that can be leveraged by those seeking to conduct research or policy related work in this area.  

Overall, the paper is very well organized and clearly written.  Areas where clarity could be improved are detailed below:

Table 2:  Only parents are listed as a significant social relationship for the emerging adulthood group, vs. family for the other age groups.  Is there a reason for excluding the broader family category here?

Thank you for this suggestion. We have revised Table 2 to include the broader family category here.

Para spanning lines 165-172 re gender roles and sexuality: The links between gender, sexuality, caregiving for intimate partners and divorce among cancer survivors are muddy.  Suggest reworking this para to make more cohesive and the potential links between findings from prior literature more clear.

Thank you for the opportunity and suggestion to revise this paragraph. This section now reads:

“The increased flexibility in gender norms within younger generations may suggest increasing gender equality in caregiving burden. For example, more young men are taking on caregiving roles than ever before [2], while simultaneously more young women work outside the home. Meaning that, the caregiving role historically taken on by women may no longer be as gendered as in the past [3,21]. In addition to less gendered norms of caregiving roles, YACCs’ gender and sexuality may be affected by caregiving during young adulthood. YACC have described their caregiving interrupting their ability to date and explore intimate relationships. Further, young adults who have survived cancer have higher rates of divorce than individuals without a cancer history [22]. While we are aware of no prior studies on the area, cancer caregiving at a young age, particularly for a spouse or intimate partner, may negatively impact these relationships.”

Line 200:  More appropriate to indicate that these impacts MAY increase the likelihood that YACCs will have to rely on public assistance in the future

Edited.

Line 208:  Consider adding the word other -- Greater physical strain has been reported in OTHER young adult caregiving groups including...

Edited.

Line 209: The term sandwich care partner is introduced here and occurs elsewhere in the manuscript but it is not defined.  This will be an unfamiliar term for some and should be clearly defined.

Thank you for this suggestion. We have defined this term at its initial use for consistency and ease of understanding. This now reads:

“Greater physical strain has been reported in other young adult caregiving groups including caregivers who provide care for young children and aging parent or grand-parents, known as sandwich caregivers (18%) and mental health care partners (22%) [16].”

Table 3:
Outcomes described for morbidity/mortality are depicted as percentages. Outcomes described for alcohol/substance use are numbers. It is hard to have a sense of the magnitude of issues when only numbers are presented. Is 4.8 million adults a large proportion of young adults? What about 3.2 million? 

Thank you for this suggestion. We have revised these numbers to be reported as percentages for ease of interpretation. This table now reads:

13.5% of young adults aged 18–25 drank alcohol an average of 4.1 drinks per day on 7 days per month, and 9.0% used marijuana [14].

Line 216 vs Table 3:
Line 216 notes that 6 of the 10 top health conditions faced by young adults include behavioral health. 
Table 3 indicates that the top 10 conditions affecting millennial health were behavioral.
Same reference provided for both statements.  Please revise for accuracy and consistency.

Thank you for noting this inconsistency. This typo has been edited to clarify that 6 of the top 10 conditions are behavioral.

Sections 9 and 10 (Research & Clinical Priorities and Summary)
Specific suggestions regarding the types of studies that may provide the most value (section 9) and the types of policy activities that could be undertaken to support YACC would strengthen the paper.

Thank you for the opportunity to revise and strengthen these recommendations. We have added the following language to these sections:

Section 9:

Future priorities for research to characterize the impact of young adult cancer caregiving include, but are not limited to, the impact of caregiving on the developmental life course trajectory, family and relationships (e.g., marriage/divorce, gender norms and equity in caregiving, childbearing), financial outcomes related to caregiving as a young adult and its impact on employment, health insurance, and lifetime earning/debt, health outcomes related to caregiving that may be amplified among young adults such as use of alcohol and other substances, negative (depression, anxiety, stress) and positive (resilience, in-dependence) psychological outcomes of caregiving, geographic considerations for long-distance caregiving, and supportive and palliative care needs among YACC who may be less familiar with severe illness.

Section 10:

Advancing policy for paid family leave, paid caregiver leave, employment and health insurance protections, and childcare support for young caregivers may support YACCs in their pursuit of personal, professional, financial, and care partner goals is critical to their overall well-being.

Reviewer 2 Report

Thank you for the opportunity to review this manuscript.

This was an interesting and well written document, which addresses an important topic. Rarely when talking about family caregivers or informal caregivers of people with cancer, attention is focused on young people and it goes unnoticed that the ages of the caregivers, the time of life they are in, determines a type of specific needs . In this sense, the contribution of this work seems very valuable.

I consider that it is a theoretical work that addresses in depth and in a very complete way the different aspects and situations that are affected due to the dedication to the care of a person (almost always emotionally significant for the caregiver) sick with cancer.

However, I have to tell you that in order to improve your manuscript you should specify in the title or at least in the abstract that it is a theoretical article, not an investigation, so that the reader can adjust the expectations raised by a subject such as the they deal with in their work.

I congratulate you for having addressed this topic in depth and providing very valuable information that should be shared by others.

Author Response

Reviewer 2

Thank you for the opportunity to review this manuscript.

This was an interesting and well written document, which addresses an important topic. Rarely when talking about family caregivers or informal caregivers of people with cancer, attention is focused on young people and it goes unnoticed that the ages of the caregivers, the time of life they are in, determines a type of specific needs . In this sense, the contribution of this work seems very valuable.

I consider that it is a theoretical work that addresses in depth and in a very complete way the different aspects and situations that are affected due to the dedication to the care of a person (almost always emotionally significant for the caregiver) sick with cancer.

However, I have to tell you that in order to improve your manuscript you should specify in the title or at least in the abstract that it is a theoretical article, not an investigation, so that the reader can adjust the expectations raised by a subject such as the they deal with in their work.

I congratulate you for having addressed this topic in depth and providing very valuable information that should be shared by others.

Thank you so much for your appreciation and enthusiasm for this work. We have revised the abstract and title to clarify the content.

The title now reads: Young adult cancer care partners: A theoretical description of an emerging population with unique needs

The abstract now reads: Herein we demonstrate through a theoretical description that young adult cancer care partners deserve distinct recognition in the cancer control continuum given the psychological, physical, financial, and social features unique to their cancer experience.